# A Radiomic-Based Machine Learning Model Predicts Endometrial Cancer Recurrence Using Preoperative CT Radiomic Features: A Pilot Study

**DOI:** 10.3390/cancers15184534

**Published:** 2023-09-13

**Authors:** Camelia Alexandra Coada, Miriam Santoro, Vladislav Zybin, Marco Di Stanislao, Giulia Paolani, Cecilia Modolon, Stella Di Costanzo, Lucia Genovesi, Marco Tesei, Antonio De Leo, Gloria Ravegnini, Dario De Biase, Alessio Giuseppe Morganti, Luigi Lovato, Pierandrea De Iaco, Lidia Strigari, Anna Myriam Perrone

**Affiliations:** 1Department of Medical and Surgical Sciences, University of Bologna, 40126 Bologna, Italy; camelia.coada@unibo.it (C.A.C.); marco.distanislao@studio.unibo.it (M.D.S.); lucia.genovesi@studio.unibo.it (L.G.); antonio.deleo@unibo.it (A.D.L.); myriam.perrone@unibo.it (A.M.P.); 2Department of Medical Physics, IRCCS Azienda Ospedaliero-Universitaria di Bologna, 40138 Bologna, Italy; miriam.santoro@studio.unibo.it (M.S.); giulia.paolani@studio.unibo.it (G.P.); lidia.strigari@aosp.bo.it (L.S.); 3Pediatric and Adult CardioThoracic and Vascular, Oncohematologic and Emergency Radiology Unit, IRCCS Azienda Ospedaliero-Universitaria di Bologna, 40138 Bologna, Italy; vladislav.zybin@studio.unibo.it (V.Z.); cecilia.modolon@aosp.bo.it (C.M.); luigi.lovato@aosp.bo.it (L.L.); 4Division of Oncologic Gynecology, IRCCS Azienda Ospedaliero-Universitaria di Bologna, 40138 Bologna, Italy; stella.dicostanzo@unibo.it (S.D.C.); marco.tesei@aosp.bo.it (M.T.); 5Solid Tumor Molecular Pathology Laboratory, IRCCS Azienda Ospedaliero-Universitaria di Bologna, 40138 Bologna, Italy; dario.debiase@unibo.it; 6Department of Pharmacy and Biotechnology, University of Bologna, 40126 Bologna, Italy; gloria.ravegnini2@unibo.it; 7Radiation Oncology, IRCCS Azienda Ospedaliero-Universitaria di Bologna, 40138 Bologna, Italy; alessio.morganti2@unibo.it

**Keywords:** endometrial cancer, pre-surgical risk, recurrence, prognosis, radiomic, artificial intelligence, personalized medicine

## Abstract

**Simple Summary:**

Accurate prediction of the risk of endometrial cancer (EC) recurrence is crucial to identify the best treatment and achieve the most favorable outcome. Currently, no model is available to predict this recurrence risk using pre-surgical computed tomography (CT) scans. This pilot study was carried out to investigate the potential of radiomic features extracted from CT scans to accurately predict the risk recurrence in such patients. The results showed that a machine learning-based model trained on CT radiomic features was able to predict EC recurrence risk with high accuracy. These results suggest that radiomics analysis using pre-surgical CT scans may provide a valuable tool for predicting recurrences in patients with EC. Further independent studies are required to strengthen these findings.

**Abstract:**

Background: Current prognostic models lack the use of pre-operative CT images to predict recurrence in endometrial cancer (EC) patients. Our study aimed to investigate the potential of radiomic features extracted from pre-surgical CT scans to accurately predict disease-free survival (DFS) among EC patients. Methods: Contrast-Enhanced CT (CE-CT) scans from 81 EC cases were used to extract the radiomic features from semi-automatically contoured volumes of interest. We employed a 10-fold cross-validation approach with a 6:4 training to test set and utilized data augmentation and balancing techniques. Univariate analysis was applied for feature reduction leading to the development of three distinct machine learning (ML) models for the prediction of DFS: LASSO-Cox, CoxBoost and Random Forest (RFsrc). Results: In the training set, the ML models demonstrated AUCs ranging from 0.92 to 0.93, sensitivities from 0.96 to 1.00 and specificities from 0.77 to 0.89. In the test set, AUCs ranged from 0.86 to 0.90, sensitivities from 0.89 to 1.00 and specificities from 0.73 to 0.90. Patients classified as having a high recurrence risk prediction by ML models exhibited significantly worse DSF (*p*-value < 0.001) across all models. Conclusions: Our findings demonstrate the potential of radiomics in predicting EC recurrence. While further validation studies are needed, our results underscore the promising role of radiomics in forecasting EC outcomes.

## 1. Introduction

Endometrial cancer (EC) has become the most prevalent gynecological malignancy with rising incidence and mortality rates, particularly in high-income countries [1]. This alarming trend has been primarily linked to the escalation of obesity rates, alongside other predisposing factors such as physical inactivity and metabolic syndrome [2].

In general, early stages of low-grade endometrioid ECs are associated with a favorable prognosis, whereas high-grade endometrioid and other types of ECs are characterized by a high risk of recurrences and a poor prognosis. Even though this latter group represents a small proportion of these patients (approximatively 20%), half of all EC-related deaths are found in this category [3,4]. Nevertheless, some low-grade tumors may also exhibit aggressive behavior. Therefore, the need to better stratify EC patients with high-risk of recurrence and death has continuously challenged the various classification systems of EC, starting from the Bokhman histologically-based classification to the newer, more complex, molecular-based ones [5,6]. Currently, the European Society of Gynecological Oncology (ESGO)/European Society for Radiotherapy and Oncology (ESTRO)/European Society of Pathology (ESP) guidelines recommend the integration of multiple layers of information including clinical, histological and molecular features [7]. However, their primary limitation is the timing of data acquisition, which occurs after surgery. Additionally, generalizing results based on a few histological or molecular markers is a significant concern. For instance, while the current classification system places significant value in stratifying the high-grade pathogenic somatic mutations in the exonuclease domain of the replicative DNA polymerase epsilon (*POLE*) mutated (with good prognosis) from P53 abnormal (with poor prognosis) for the selection of adjuvant treatment [7,8], it has limited usefulness on the intermediate prognosis group (no specific molecular profile (NSMP) and microsatellite instability (MSI) molecular subtypes).

In recent years, the radiomics field has gained significant momentum, with an ever-growing body of evidence suggesting its potential and capacity in identifying features that are otherwise unseen by the human eye, as well as their association with various clinical and pathological features [9]. In EC, pre-surgical radiological assessment is employed for disease staging, particularly for the evaluation of lymph node involvement, which is required for the selection of the appropriate treatment. However, this approach can underestimate the degree of disease extension, as important prognosis-associated features such as myometrial invasion and lymphovascular invasion (LVI), are only identified by the pathologist in post-surgical histological samples [4,10]. Selecting patients by noninvasive methods at diagnosis to tailor the most appropriate surgical strategy has always been the clinician’s goal, in order to reduce morbidity and mortality risks associated with overtreatment. The integration of radiomics with the imaging modalities already employed in clinical practice could lead to a better preoperative evaluation and more efficient management [11].

To date, various studies have explored the usefulness of radiomics-based artificial intelligence (AI) models in identifying different features of EC associated with lymph node metastases [12,13,14], LVI [12,15,16], as well as assisting in the differential diagnosis of EC from benign lesions [17,18] with promising results. Most of the studies have used magnetic resonance imaging (MRI) as input for feature extraction. However, limited research is available regarding the applicability of radiomic models on Computer Tomography (CT) scans. Despite MRI being the recommended imaging technique for the pelvic region, pre-surgical cancer staging is performed using CT scans worldwide due to their wider distribution and faster acquisition times. So far, CT-based radiomics coupled with machine learning (ML) models have demonstrated promising results in predicting the progression-free survival of other tumor types [19] including small cell lung cancer [20], esophageal cancer [21], glottic cancer [22] and gastric cancer [23]. However, such analyses have not been carried out yet for EC.

Thus, the aim of this pilot study was to develop and test a non-invasive AI-based model capable of predicting relapse of EC patients by utilizing radiomic data extracted from presurgical CT scans.

## 2. Materials and Methods

### 2.1. Study Cohort and Clinicopathologic Data

This retrospective, observational study was approved by the local Ethical Committee with the number 189/2021/Oss/AOUBo. All patients underwent clinical and pathological diagnosis, surgical and pathological staging, and follow-up at the Division of Oncologic Gynecology IRCCS Azienda Ospedaliero-Universitaria di Bologna, Bologna, Italy [24]. Electronic medical files were used to obtain all patients’ data, including age at diagnosis, Body Mass Index (BMI), International Federation of Gynecology and Obstetrics (FIGO) stage determined using surgical and pathological reports, European Society of Medical Oncology (ESMO) 2016 risk stratification group, type of surgery, peri-operative complications, imaging studies and pathology reports. The pathological report and stage were obtained following the ESGO/ESTRO/ESMO guidelines [25,26]. The Cancer Genome Atlas Program (TCGA)/The Proactive Molecular Risk Classifier for Endometrial Cancer (PROMISE)/Post Operative Radiation Therapy in Endometrial Carcinoma (PORTEC) analysis was performed as previously described [27,28,29,30] (Appendix A).

### 2.2. Patients CT Scans, Image Selection and Contouring

Pre-surgical CT scans, routinely performed in our unit for EC disease staging, were extracted from the Picture Archiving and Communication System (PACS) of our hospital. We excluded CT scans in which image artifacts due to hip prosthesis, or movement artifacts were detected.

The CT examinations were performed on different CT scanners, i.e., LightSpeed 16, LightSpeed VCT, Discovery CT 750, Optima 660 by GE Healthcare (GE Healthcare, Milwaukee, WI, USA), Brilliance iCT 128, Ingenuity, Brilliance 16 by Philips (Philips Healthcare, Cleveland, OH, USA), and SOMATOM Definition Edge, Emotion 6 by Siemens (Siemens Healthcare, Forcheim, Germany). Overall, the CT images were acquired with the following parameters: mA range: 23–605, kV range: 100–140, helical technique: helix; slice thickness range: 1–5 mm; IV administered low-osmolality non-ionic iodinated contrast agent dose: 90–150 mL. All the images were reconstructed with a soft tissue algorithm and aligned to the body axis. The radiomic analysis utilized the venous phase from Contrast-Enhanced CT (CE-CT) scans. This phase has been noted to be more valuable for endometrial cancer staging compared to the arterial phase as it particularly highlights parenchymal characteristics, contrast dynamics, and the contrast between the tumor and myometrium [31,32,33,34,35,36]. Both radiologists and gynecology experts blind to the pathological results performed a slice-by-slice evaluation of the CT scans and semi-automatically contoured the complete volume of each patient’s uterus using the MIM software (v. 7.1.4, MIM Software Inc. Cleveland, OH, USA) to obtain the volumes of interest (VOIs). Particular attention was paid to ensure the exclusion of surrounding intestinal and vascular structures. The inter-physician reproducibility was assessed on a subset of patients through the evaluation of the Dice Similarity Coefficient (DSC) and the Mean Distance to Agreement (MDA) which are the most common measures in terms of geometric quantification of contour similarities [37]. The DSC serves as both a spatial overlap index and a metric for validating reproducibility. Its value varies between 0 and 1, representing the extent of spatial overlap between two sets of binary segmentation results. A DSC of 0 signifies no spatial overlap, while a DSC of 1 indicates complete overlap [38]. The MDA describes the mean voxel-wise comparison of the distance between two associative points in two contour sets [39], thus representing a measure of average similarity between them. In particular, an MDA of 0 indicates that the two contours are identical. In this study, DSCs ≥ 0.8 and MDA values ≤ 3 mm were used to verify the agreement among the physicians, according to the study by Strolin et al. [40].

### 2.3. Radiomic Features Extraction and Selection

After the contouring step, the resulting VOIs were used for the feature extraction with an ad hoc developed Python (v. 3.8.3) [41] script including the PyRadiomics library [42]. In total, 107 radiomic features (RFs) were calculated from eight different classes: First Order Statistics, 3D Shape-based, 2D Shape-based, Gray Level Cooccurrence Matrix (GLCM), Gray Level Run Length Matrix (GLRLM), Gray Level Size Zone Matrix (GLSZM), Neighboring Gray Tone Difference Matrix (NGTDM) and Gray Level Dependence Matrix (GLDM). Prior to the RFs extraction, a pre-processing operation was performed and implemented in the Python script since the CT images were acquired with different scanners and protocols. Specifically, the images were resampled to obtain an isotropic voxel spacing of 5 mm and intensity gray level discretization in 64 bins.

### 2.4. Construction of the Radiomic Predictive Model

The machine learning (ML) model was created using RStudio v.1.2.1335 (R Core Team, Vienna, Austria), with coding performed in R v.4.1.3 language (https://www.r-project.org/ accessed on 23 December 2022). The database of RFs was used for developing three predictive classifiers with ML-based supervised approaches having set the EC recurrence and the time to the recurrence event as outputs. Figure 1 shows the study design in a schematic way.

In particular, the 107 RFs were scaled using the z-score, then, by generating random seeds to make the results repeatable, the dataset was divided with a balanced output (i.e., disease recurrence) in 60% training and 40% testing 100 times. After the automatic feature extraction, the patients were divided into training and test sets with a 60:40 ratio. The training set consisted of 49 EC patients, of which nine (18.4%) presented disease recurrence during follow-up, while the test set included the remaining 32 patients out of which six (18.7%) presented recurrence during the follow-up period. After the application of the data augmentation and balancing operations, the training data consisted of 318 EC patients of which 158 had recurrence (49.7%). The augmented and balanced dataset was created using the *smoteclassif* function from UBL R-package [43] which implements the SMOTE algorithms [44,45], adopting a number of nearest neighbors (i.e., k parameter) equal to 5 (corresponding to the default value), balancing the ratio of 4.4 between majority and minority classes and multiplying by an augmentation factor of 4. This value was chosen to obtain the minimum number of patients (i.e., 300) required to create the ML-based model [46]. In each augmented and balanced training dataset, a Kaplan-Mayer-based univariate analysis with a *p*-value ≤ 0.2 was performed on the initial RFs to reduce the dimensionality of the database.

All the remaining RFs were included in three different ML-models for the disease recurrence-free prediction: the *LASSO-Cox* model, i.e., a Cox proportional-hazards regression regularized using the Least Absolute Shrinkage and Selection Operator (LASSO), the *CoxBoost* model [37], i.e., a Cox model with a likelihood-based boosting, and the *RFsrc* model, i.e., a random forest model optimized for the survival analysis. In the LASSO-Cox model, the LASSO penalization method shrinks the coefficients of non-relevant RFs to zero, considering both the binary output and the variable of the time frame up to the event (i.e., the months up to the recurrence event) simultaneously. Regarding the choice of the best hyperparameter (i.e., the penalty term λ), the *cv.glmnet* function facilitated the performance of a 10-fold CV to find the minimum λ [47]. This hyperparameter was used in the *glmnet* function to create the investigated LASSO-Cox models [48,49,50]. On the other hand, in the *CoxBoost* model, implemented in R through the *CoxBoost* function [51], the Cox regression coefficients are updated at each iteration and a “weak” estimator is fitted to the modified version of the data to minimize a pre-specified loss function. The final estimate is given by the combination of all the small contributions used to update the parameters estimation [51]. In this case, for the choice of the best hyperparameter (i.e., the *boosting step size,* which controls the weakness of the estimator), the *cv.CoxBoost* function was used to perform a 10-fold CV, thus finding the optimal number of boosting steps, i.e., the one having the minimum mean partial log-likelihood. Finally, the RFsrc model, implemented in R through the RFsrc library, allows for the extension of the random forest approach to the survival analysis [52]. In this case, the RFs were selected using the variable hunting approach in the *var.select* function and the top variables were included in the RFsrc model.

For each of the ten iterations, the prediction was performed on both groups by extracting the Receiver Operating Characteristic (ROC) curve with Area Under the Curve (AUC), sensitivity, specificity, and confidence interval. The threshold value of the ROC curve obtained for the training dataset calculated with the Youden’s criterion was used also in the test dataset. For each approach, models with a statistically significant AUC (i.e., with a confidence interval between 0.5 and 1) in both training and test sets were selected and the Bayesian information criterion (BIC) was used to identify the optimal model [53], while the highest AUC was used for the RFsrc-based model.

Finally, the Kaplan-Meier method was implemented to show the survival curves using the previously defined ROC threshold.

### 2.5. Statistical Analysis

We assumed a threshold probability for rejecting the null hypothesis α (two-tailed) = 0.05, a power of 80%, a proportion of RF-based high-risk subjects of 0.25 and a relative hazard (high-risk group/low-risk group) = 0.15, thus, the total events needed are 12. Considering a baseline event rate (events/unit time) for the low-risk group = 0.35, a median survival time in the groups = 1.5 years, a censoring rate—censored/unit time (assumed equal for both groups) = 0 and an average length of follow-up FU = 1.5 years, we calculated a sample size of 60 patients [54].

Descriptive statistics were presented as absolute values and percentages for categorical variables and mean ± standard deviation (SD) for continuous variables. The chi-square test was used to analyze differences in the distribution for the categorical variables, while the *t*-test was used for continuous ones.

ROC curves and AUC, representing the correct classification probabilities, were used to calculate the ability of the radiomic classifier to predict the recurrence of EC after primary treatment. Disease-Free Survival (DFS) curves were built using the Kaplan-Meier estimator and the log-rank test was used to assess the statistical significance. The reverse Kaplan-Meier method was used to calculate the overall follow-up time of patients.

Analyses were performed using SPSS statistical software v26 (SPSS Inc. IBM, Chicago, IL, USA) and R v.4.1.3, using the following packages: pROC, survminer. The significance threshold was set at a *p*-value of 0.05.

## 3. Results

### 3.1. Patient Characteristics

The study included a total of 81 patients who were diagnosed with EC at the Division of Oncologic Gynecology, IRCCS Azienda Ospedaliero-Universitaria di Bologna, Italy. The characteristics of the patients are summarized in Table 1. Most patients were diagnosed at an early stage, namely 61 (75.3%) at stage I, six (7.4%) at stage II, and 14 (17.3%) at stage III. Endometrioid EC was the most frequently observed histotype (62, 76.5% of patients), followed by serous (nine; 11.1% of patients) and the dedifferentiated subtypes (nine; 11.1% of patients). No significant differences were observed between patients with and without recurrent disease in terms of age at diagnosis, BMI, associated comorbidities such as hypertension and diabetes, familiar history of cancer, age of menarche and menopause, and the use of hormonal therapy (*p*-values > 0.05 for all variables; Table 1).

The median follow-up time was 32.27 months (interquartile range 10.23; 55.3). During this period, a total of 15 (18.5%) patients experienced recurrence, and 13 (16%) patients died of disease. The main characteristics of the patients with recurrent EC were: endometroid histotype (eight; 53.3%), high-grade (11; 73.3%), P53 abnormal molecular subtype (seven; 46.6%) and were classified as having high-intermediate and high ESMO risk (two; 13.3% and 13; 86.7%, respectively). None of the low-grade EC patients in this study experienced recurrent events. The most common sites of recurrence were lymph nodes (five; 33.3%) and distant sites such as brain and lungs (five; 33.3%) followed by local peritoneal (four; 26.7%) and vaginal site (one; 6.7%) recurrences.

### 3.2. Radiomic Model Prediction of Recurrence

The uterine bodies of all 81 pre-surgical patients were contoured in a semi-automatic manner by a gynecological surgeon and a radiologist. The time between CT and surgery was less than two months for all analyzed patients. All images were cross-checked and corrected as necessary. The VOIs drawing process showed a rapid learning curve, with the mean initial time required for contouring of approximately 20–25 min/patient for the first ten patients, which then decreased to approximately 10 min/patient. In a subset of 18 patients (about 22%) from the investigated cohort, the MDA values were below the chosen threshold of 3 mm, while the DSC values exceeded the value of 0.8 in 83% of the total cases, representing a good inter-physician reproducibility. Figure 2 shows examples of contoured VOIs in EC patients with and without disease recurrences, along with the 3D reconstructions.

Table 2 shows the AUC (95% CI), sensitivity and specificity of the optimal ML-based models selected on training set according to the BIC for the LASSO-Cox and CoxBoost approaches and to the highest AUC value on training set for the RFsrc approach. The table also reports the threshold obtained from the ROC curve on the training set belonging to the best ML-based models (i.e., 0.61, 0.55 and 57.01 for LASSO-Cox, CoxBoost and RFsrc models, respectively). Figure 3 shows the Kaplan-Meier curves for the DFS obtained using the above thresholds on test set (i.e., patients non used to train the model) with the corresponding the log-rank tests, the *p*-values were < 0.001 from all the optimal ML-based models. These thresholds were used to build the ROC curve on the test set and to divide patients into high-risk and low-risk recurrence groups (pred_high_risk and pred_low_risk) (Figure 3). The obtained recurrence risk label was assigned to EC patients and used to build the Kaplan-Meier curves to estimate the recurrence probabilities. As shown in Figure 3, the log-rank tests were significant for the DFS on test set (i.e., patients not used to train the model) with a *p*-value < 0.001 obtained from all the optimal ML-based models. In the testing group, the median DFS for the EC patients classified as having a high-risk for recurrence by the radiomic predictive models was 22.4 months, 25.9 months and 14.1 months for RFsrc, CoxBoost and LASSOCox, respectively.

Table 3 shows the RFs selected by the optimal ML-based models reported in Table 2 and their importance, i.e., their degree of usefulness/contribution to the model. Of note, for the RFsrc model, we reported the depth of the features as well as their frequency of occurrence in five repeated iterations. Specifically, the feature depth refers to how deep a variable is used for splits in a decision tree: variables near the tree’s root capture broad patterns, while deeper variables capture finer details. At the same time, the feature occurrence indicates how often a RF is chosen for splitting across all trees in the forest: high occurrence shows strong influence on predictions, while low occurrence suggests less impact. In essence, the depth of a RF affects its contribution at different decision levels, while its occurrence indicates its importance across the entire forest. These factors collectively shape how random forest trees make predictions. Overall, most of the RFs detected by the optimal ML-based classifiers belonged to higher order classes. Exceptions are represented by only four (i.e., 10Percentile, Kurtosis, Minimum, and Variance) and three (i.e., Flatness, LeastAxisLength, and Maximum2DDiameterColumn) RFs, which belonged to first-order and shape classes, respectively.

Finally, the DFS Kaplan-Meier curves based on molecular classes did not provide a significant result in our patient cohort (*p* = 0.12, Appendix A) while the ones based on their binary classification (i.e., P53 versus the other molecular classes) showed a significant trend (*p* = 0.03 Appendix A).

## 4. Discussion

To the best of our knowledge, this is the first study to evaluate the accuracy of CE-CT-based radiomic features in predicting recurrence in EC patients, highlighting its potential as a valuable decision support tool in clinical settings. The results of our study provide convincing evidence of the potential exhibited by ML approaches applied to radiomics data for predicting DFS.

Imaging techniques are crucial in assessing EC patients, but their availability varies across countries. In settings with ample resources, MRI is the preferred modality for investigating the pelvic region and identifying the infiltration and extension of EC [55]. However, low-income countries may lack access to MRI [56]. Positron Emission Tomography (PET) imaging has been introduced for staging and prognosis [57], but its performance is limited in detecting lymph node metastases [58]. However, combining high-resolution diagnostic images with radiomics showed promise in predicting lymph node metastases [13,59,60]. CTs, on the other hand, are more widely available and provide satisfactory staging but have limited usefulness compared to MRI and PET in EC [55]. Thus, ML models using CT scans could provide prognostic information in medical centers with fewer resources [61], offering personalized care for EC patients, and improving pre-surgical risk assessment. Our study’s ML models provide prognostic information using noninvasive CE-CT images before surgery. Given that around 18.5% of early-stage cases encounter relapse, the imperative significance of precisely identifying patients for optimal treatment selection cannot be overstated [4]. Thus, various classifications aim to identify risk factors to guide surgical interventions and adjuvant therapies [3,62]. However, this information is usually obtained after primary surgery, making it challenging to adjust its extent [63]. Our ML models can predict EC recurrence, which could aid ultra-staging and help surgical teams determine the best approach for each patient. Additionally, our ML-based models outperform other predictors (i.e., the TCGA molecular classification of EC) and are independent of postoperative parameters.

Currently, the role of CT is generally restricted to the preoperative evaluation of the disease burden in relation to extra-uterine spread, lymphadenopathy and metastatic disease beyond the pelvis, given the low sensitivity (83%) and low specificity (42%) of CT in assessing myometrial involvement and cervical stromal invasion [56,64]. To date, most radiomic analyses have been conducted using MRI data, with only a few studies focusing on the value of CT scans. For example, radiomic data extracted from PET/CT images and/or combined with PET/CT-based parameters have been shown to be useful for the detection of nodal metastases and staging of EC tumors [60,65] while CE-CT has proven useful for the identification of microsatellite instability [66]. In terms of DFS in EC, few studies have been published. Nakajo et al. created a [^18^F]-FDG PET-based radiomic ML model and obtained an AUC of 0.89 for the prediction of PFS [59], while other studies using MRI-based models alone or combined with clinical data obtained mean AUCs varying between 0.62 and 0.85 [67,68,69]. To complete this knowledge gap, our study aimed to evaluate the efficacy of CT-based radiomics in predicting recurrences in EC. We used the radiomic data extracted from the venous phase of CE-CT images to train and establish three ML-based algorithms. When applying the algorithms on the EC testing group, we obtained an optimal stratification of high-risk EC patients with accuracy, sensitivity, and specificity ranging from 0.78 to 0.9, from 0.89 to 1.00, and from 0.73 to 0.90, respectively. Hence, our study lays the ground for a potential resource for the identification of EC patients prone to relapse, which could aid in the development of efficacious prevention and management strategies.

All three models resulted in having an equivalent predictive ability and allowed us to identify the most important RFs, which mostly belonged to high order classes. It is worth noting that having multiple concurring models helps increase the robustness and reliability of the results. Each model utilizes different algorithms or methodologies, thereby reducing the risk of any single model’s biases or limitations influencing the obtained results. When multiple models agree on a particular prediction or finding, it enhances confidence in the validity of the result. The LASSO-Cox model combines the LASSO (Least Absolute Shrinkage and Selection Operator) method, which performs variable selection, with the Cox proportional hazards model, which estimates the hazard function [51]. This combined approach can pinpoint the most pertinent variables and estimate their impact on the outcome. The CoxBoost approaches, on the other hand, employ boosting techniques to enhance the prediction accuracy of the Cox model [47,49,50]. Boosting iteratively combines weak learners to create a strong learner, which improves the model’s performance and reduces bias. Lastly, the RFsrc-based model utilizes the random forest algorithm [52], which leverages the power of ensemble learning. It constructs multiple decision trees and combines their predictions to produce a robust and accurate model. Random forests can handle complex interactions between variables and provide insights into feature importance. By employing these three concordant models, we were able to leverage the unique strengths of each approach, mitigating the weaknesses, and gaining a comprehensive understanding of the data. This complex approach using different models represents a strength of our work despite the relatively small number of patients used in this pilot study. Another strength of our study is represented by the fact that our models provide important prognostic information before any therapeutic approach, which was not previously possible.

It is worth discussing that the SMOTE algorithm implemented for data augmentation and balancing could be affected by an over-generalization problem which could lead to the inability to capture the characteristics of a specific under-represented group of patients. This drawback can be caused by the possible oversampling of noisy or uninformative samples or the increasing overlap between different classes around the class boundaries [70]. Nevertheless, beyond the internal validation performed using a CV approach as suggested by Papanikolaou et al. [71], further studies conducted on wider cohorts are needed to externally validate these results and identify radiomic markers that can be standardized and easily applied on CE-CT images collected in clinical practice in all medical centers. External validation was performed only in a limited number of studies, i.e., about an average of 6% of the cases according to a review evaluating 516 models [72] and could be further complicated by the lack of standardization among different scan machines and protocols implemented in the involved hospitals, thus affecting the radiomic pipeline and requiring a harmonization procedure before the feature extraction in multicentric studies [73]. To better generalize the proposed ML-based models, our institute is starting a prospective multicentric study based on CT-based RF and DFS in EC patients. This paves the way for a possible future validation and adoption as a valuable ML-based tool in personalized medicine for EC patients.

## 5. Conclusions

Our study showed that radiomic features from pre-surgical CE-CT scan of EC tumors can be a valuable tool for predicting recurrences in these patients. By stratifying patients based on the developed models, surgical and oncological teams can make more informed decisions about the most suitable treatment options, ultimately improving patient outcomes. Compared to the molecular stratification, which often requires specialized laboratories and costly tests, such as identification of *POLE* mutations, CT-based models can be more widely adopted, even in hospitals with limited access to MRI and PET/CT resources. This represents a significant step towards the implementation of more personalized medicine and has the potential to greatly benefit EC patients and healthcare providers alike.

## Figures and Tables

**Figure 1 cancers-15-04534-f001:**
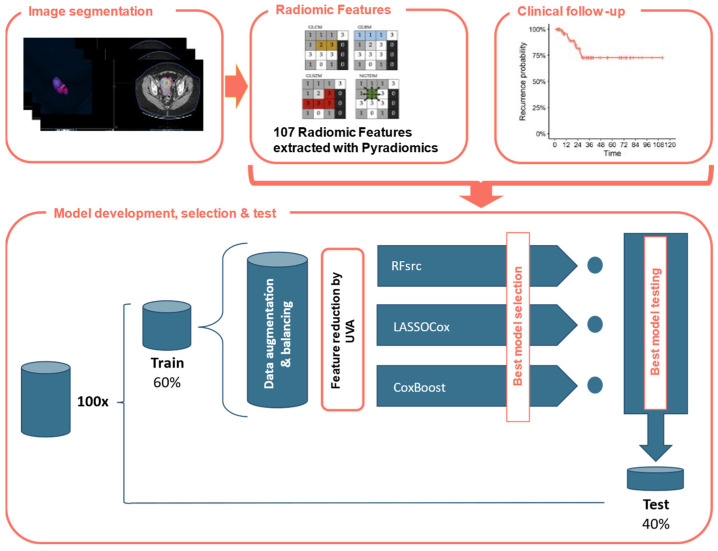
Study design. CE-CT images were delineated, and Radiomic Features (RFs) were automatically extracted using Pyradiomics. Each patient was associated with observed disease-free survival. The dataset was divided into training and test sets with a 60:40 ratio 100 times. After the application of the data augmentation and balancing operations, the training data consisted of 318 EC patients of which 158 had recurrence (49.7%). A RFs reduction was performed using Kaplan and Meier curves and calculating the *p*-value of log-rank test assuming a cutoff of 0.20. Thus, the optimal model was applied using the following ML-based models: LASSO-Cox, CoxBoost and RFsrc.

**Figure 2 cancers-15-04534-f002:**
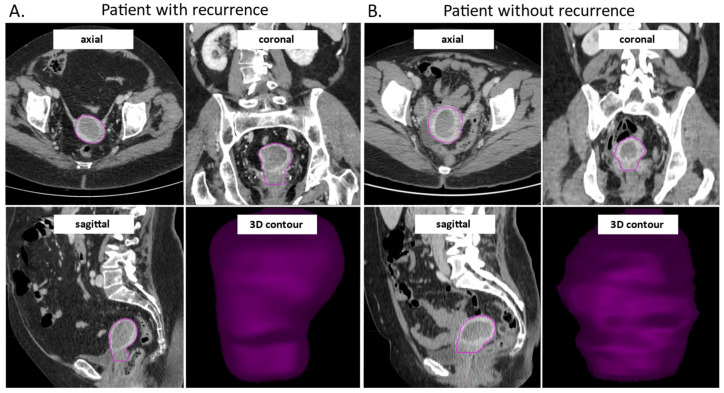
CE-CT images in axial, coronal and sagittal view, and the 3D visualization of the contoured VOI from a patient with early EC recurrence (**A**) and without early recurrence during study follow-up (**B**). The images were acquired in the venous phase of contrast enhancement. CE-CT: contrast enhanced computer tomography, VOI: volume of interest; EC: endometrial cancer.

**Figure 3 cancers-15-04534-f003:**
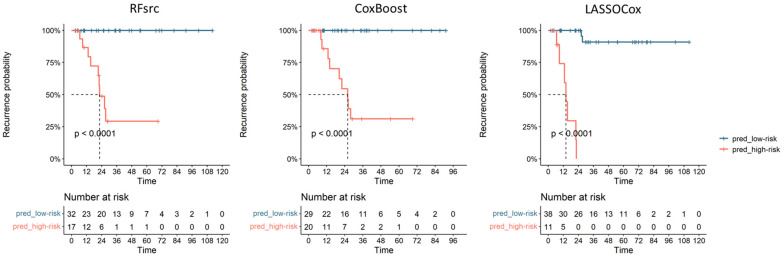
Performance of the radiomic models on the test EC cohort with the Kaplan-Meier curves showing patients’ recurrence based on the three ML models’ predictions; pred_low-risk and pred_high-risk: ML predictive scores for recurrence.

**Table 1 cancers-15-04534-t001:** Clinical and pathological characteristics of EC patients included in the study. Summary statistics are presented as counts (percentages) or mean ± standard deviation. N: number of patients; N/A: not available; BMI: body mass index; FIGO: Federation Internationale de Gynecolgie et d’Obstetrique; BRT: brachytherapy; RTE: external radiotherapy; CT: chemotherapy; w/o: without; w/: with.

			No Recurrence N = 66	With Recurrence N = 15	*p*-Value
Age at diagnosis (years) mean ± SD			60.7 ± 11.4	65.9 ± 8.6	0.1
BMI N (%) *N/A* = 1			28.4 ± 7.3	26.7 ± 4.9	0.39
Associated comorbidities N (%)	Hypertension *N/A* = 15	No	29 (53.7)	5 (41.7)	0.53
		Yes	25 (46.3)	7 (58.3)	
	Diabetes *N/A* = 15	No	44 (81.5)	11 (91.7)	0.67
		Yes	10 (18.5)	1 (8.3)	
Positive family cancer history N (%) *N/A* = 23		No	23 (37.7)	6 (42.9)	0.77
		Yes	38 (62.3)	8 (57.1)	
Age at menarche (years) mean ± SD			12.5 ± 1.7	12.9 ± 1.1	0.56
Cause of menopause N (%) *N/A* = 2	Physiological		52 (81.3)	13 (86.7)	1
	Iatrogenic		12 (18.8)	2 (13.3)	
Age at menopause (years) mean ± SD			49.2 ± 5.6	52.2 ± 3.5	0.1
Hormonal therapy N (%) *N/A* = 2	No		54 (84.4)	13 (86.7)	1
	Yes		10 (15.6)	2 (13.3)	
Histotype N (%)	Endometrioid		54 (81.8)	8 (53.3)	0.02
	Serous/papillary		4 (6.1)	5 (33.3)	
	Mucinous		1 (1.5)	0	
	Dedifferentiated		7 (10.6)	2 (13.4)	
Tumor Grade N (%)	G1		18 (27.3)	0	0.01
	G2		25 (37.9)	4 (26.7)	
	G3		23 (34.9)	11 (73.3)	
Molecular class N (%)	POLE		12 (18.2)	0	0.03
	MSI		24 (36.4)	4 (26.7)	
	P53		10 (15.1)	7 (46.6)	
	NSMP		20 (30.3)	4 (26.7)	
Stage (FIGO 2014) N (%)	I		58 (84.8)	5 (33.3)	<0.001
	II		3 (4.5)	3 (20)	
	III		7 (10.6)	7 (46.7)	
ESMO risk class w/o molecular class N (%)	Low		34 (51.5)	0	<0.001
	Intermediate		8 (12.1)	0	
	High-intermediate		7 (10.6)	3 (20)	
	High		17 (25.8)	12 (80)	
ESMO risk class w/molecular class N (%)	Low		40 (49.4)	0	<0.001
	Intermediate		9 (13.6)	0	
	High-intermediate		4 (6.1)	2 (13.3)	
	High		13 (19.7)	13 (86.7)	
Lymph nodal metastasis *N/A* = 13			4 (7)	6 (54.5)	0.001
Received adjuvant therapy N (%)			36 (54.4)	15 (100)	0.001
Type of adjuvant therapy N (%) *N/A* = 4	BRT		13 (38.2)	3 (20)	0.5
	RTE + BRT		2 (5.9)	1 (6.7)	
	CT		5 (14.7)	3 (20)	
	CT + RTE		7 (20.6)	2 (13.3)	
	CT + BRT		1 (2.9)	0	
	CT + RTE + BRT		6 (17.6)	6 (40)	
Recurrence site N (%)	Local pelvic peritoneal		-	4 (26.7)	n/a
	Vaginal		-	1 (6.7)	
	Lymph nodal		-	5 (33.3)	
	Distant		-	5 (33.3)	

**Table 2 cancers-15-04534-t002:** ROC parameters obtained from the optimal LASSO Cox-model for training and test set EC groups. Th: threshold; Sp: specificity; Se: sensitivity; NPV: negative predictive value; PPV: positive predictive value; 95% CI: 95% confidence intervals; AUC: area under the curve; #: number of.

					Train				Test		
Model	AICc	BIC	# RFs	Seed	AUC (95% CI)	Se	Sp	ROC Th	AUC (95% CI)	Se	Sp
RFsrc	NA	NA	2	4306	0.93 (0.91–0.96)	0.96	0.79	57.01	0.90 (0.84–0.96)	1.00	0.80
CoxBoost	587.7	649.6	17	1936	0.92 (0.88–0.95)	1.00	0.77	0.55	0.86 (0.79–0.93)	1.00	0.73
LASSOCox	−580.5	−525.6	15	1510	0.92 (0.88–0.96)	0.96	0.89	0.61	0.89 (0.78–1.00)	0.89	0.90

**Table 3 cancers-15-04534-t003:** Feature importance of the selected radiomic features of the CoxBoost and the LASSOCox models. For the RFsrc, the depth of the variables as well as their frequencies of occurrence in five repeated iterations were reported. The values represent the coefficients attributed to each feature in the ML model.

	Coefficient Values in the ML Model			
Feature Names	CoxBoost	LASSOCox	RFsrc	
			Depth	Frequency
original_firstorder_10Percentile		−1.2131		
original_firstorder_Kurtosis	−0.2445	−2.1338		
original_firstorder_Minimum		−2.0737		
original_firstorder_Variance	−1.8482			
original_glcm_ClusterShade	2.4467	−1.4246		
original_glcm_Contrast		0.8503		
original_glcm_Correlation	−0.5000			
original_glcm_Imc2	0.9104	−0.0002		
original_glcm_JointEntropy	1.4399	−4.8506		
original_glcm_MaximumProbability	−0.0274	−1.7325		
original_glcm_SumEntropy	0.0306			
original_glcm_SumSquares	1.4606			
original_gldm_DependenceNonUniformity	−0.4479			
original_gldm_HighGrayLevelEmphasis			0.4535	80
original_gldm_SmallDependenceHighGrayLevelEmphasis		−2.5705		
original_glrlm_GrayLevelNonUniformity	−0.0495			
original_glrlm_RunEntropy	−0.6193			
original_glszm_LargeAreaHighGrayLevelEmphasis		−2.2316		
original_glszm_LargeAreaLowGrayLevelEmphasis			0.6585	40
original_glszm_SizeZoneNonUniformity	−0.8672			
original_glszm_ZoneVariance	−0.0525			
original_ngtdm_Coarseness	−1.9248	−3.5406		
original_ngtdm_Complexity	−3.9808			
original_ngtdm_Contrast		1.1900		
original_shape_Flatness	1.4152	2.7535		
original_shape_LeastAxisLength		−1.8535		
original_shape_Maximum2DDiameterColumn		1.4715		

## Data Availability

The data can be shared upon reasonable request to the corresponding author.

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
