# Peer review of "A Radiomic-Based Machine Learning Model Predicts Endometrial Cancer Recurrence Using Preoperative CT Radiomic Features: A Pilot Study"

_cancers, 2023, doi:10.3390/cancers15184534_

Round 1
Reviewer 1 Report (Previous Reviewer 1)
Authors has improved their manuscript by performing data augmentation and comparing three machine learning models.
1. The main limitation is the case number (n=81). The title is therefore rephrased
by adding "a pilot study". I think it is marginally acceptable.
2. The abstract is not clear enough. For example, number of patient being enrolled.
3. Some abbreviations need to be spelled out at first use.
4. Line 96. The cited reference [21] is incorrect. Why the paper only has one page?
5. Line 127. Why only venous phase was used? How about arterial phase?
6. Line 182. Why there were 318 EC patients for taining after data augmentation?
7. Line 316. Table 3. What are the meaning of values? Please clarify.
Minor check.
Author Response
Reviewer 1
Comments and Suggestions for Authors
Authors has improved their manuscript by performing data augmentation and comparing three machine learning models.
- The main limitation is the case number (n=81). The title is therefore rephrased by adding "a pilot study". I think it is marginally acceptable.
R: Thank you for your feedback and careful consideration of our study. We appreciate your acknowledgment of the limitation in our case number (n=81), and we have taken your suggestion to rephrase the title with the inclusion of 'a pilot study' to reflect this limitation. We understand that the sample size may pose a constraint on the generalizability of our findings (we also added this to the limitation of the study). Nevertheless, our sample size estimation showed that our study would have a power of 80 to find a significant result which is a threshold commonly used and widely accepted in medical research.
We added the following sample size estimation paragraph: We assumed a threshold probability for rejecting the null hypothesis α (two-tailed) =0.05, a power of 80%, a proportion of RF-based high-risk subjects of 0.25 and a relative hazard (high-risk Group/low-risk Group) = 0.15, thus, the total events needed are 12. Considering a Baseline Event Rate (events/unit time) for low-risk Group = 0.35, a median survival time in the Groups = 1.5 years, a censoring rate - censored/unit time (assumed equal for both groups) = 0 and an average length of follow-up FU = 1.5 years we calculated a sample size of 60. (page 6, lines 233-239)
Sample size (with continuity correction) for RF-based groups:
N Events Cumulative Event Rate
Low-risk Group 45 4 0.099
High-risk Group 15 8 0.500
Total 60 12 0.199
Reference: Schoenfeld DA. Sample-size formula for the proportional-hazards regression model. Biometrics 1983;39:499-503.
With respect to highlighting that our work is a pilot study is ‘marginally acceptable’ we would like to emphasize that our study represents an initial exploration into the utilization of pre-operative CT images for predicting recurrent events in endometrial cancer patients. Given the preliminary nature of our investigation, the 'pilot study' label accurately highlights our intent to establish feasibility and identify potential trends. The radiomic approach we employed holds promise in predicting disease-free survival (DFS) among endometrial cancer patients using pre-surgical CT scans. While our case number is limited, our study aimed to demonstrate the potential value of this method and its feasibility for future research. We believe that the positive performance of our machine learning models in both training and test sets, as indicated by AUCs, sensitivities, and specificities, provides a strong foundation for further exploration and validation. We fully acknowledge the need for larger validation studies to enhance the robustness of our findings and validate the predictive potential of radiomics in EC recurrence. These encouraging results have laid the foundation for a future multicentric study we seek to perform for a further validation of our results. Once again, we appreciate your valuable feedback and will certainly consider it as we continue to refine our study and explore opportunities for future research.
- The abstract is not clear enough. For example, number of patient being enrolled.
R: Thank you for the comment. We have revised the abstract to enhance its clarity and have included the number of enrolled patients. We hope that the changes provide a clearer understanding of the study's objectives and outcomes.
- Some abbreviations need to be spelled out at first use
R: Thank you for your clarification. We have thoroughly reviewed the entire text and have corrected the spelling of all abbreviations. - Line 96. The cited reference [21] is incorrect. Why the paper only has one page?
R: Thank you again for the comment, the cited paper (“Radiomics Signature and Clinical Parameters of 18F-FDG PET/CT Predicting Progression-Free Survival in Patients with Breast Cancer: A Preliminary Study”) refers to a 18F-FDG PET/CT based radiomic model, not to a CT based radiomic model. We apologize for the inaccuracy. However, the cited paper does not have only one page. After considering this, we searched for a similar article using CT, but found none. If you concur, we propose deleting the reference.
A5. Line 127. Why only venous phase was used? How about arterial phase?
R: Our study focused on the porto-venous phases of contrast-enhanced CT scans to analyze the parenchymal phase of the uterus, particularly the 90 to 120-second interval. This approach was chosen because endometrial cancer lesions exhibit weaker enhancement compared to normal myometrium. Moreover, the endometrium is often well visualized on portal venous and delayed phases of contrast-enhanced CT and a tumor-to-myometrium contrast window of 90 to 150 seconds after contrast administration has been proposed. Additionally, Nougaret et al. suggest obtaining images at 2 minutes 30 seconds post-contrast for optimal tumor-to-myometrium contrast. Three types of intravenous uterine contrast enhancement patterns were identified, as described by Kaur et al.: type 1 with subendometrial enhancement visible between 30 to 120 seconds; type 2 with progressive external myometrial enhancement; and type 3 with mild and diffuse enhancement, observed mainly at menopause. Variable enhancement patterns can lead to confusion in multidetector CT scans. While a homogenous myometrial contrast enhancement is generally expected in fertile age women during portal venous phase CT scans, various factors like contrast administration rate and cardiac output influence enhancement patterns. Distinct patterns can transition between phases, and vascular variations contribute to differing enhancement types. Considering our focus on endometrial cancer staging, the arterial phase's relevance is limited. Our choice to include the venous phase is based on parenchymal characteristics, optimal tumor-myometrium contrast, and contrast dynamics. This decision reflects a justification and optimization of radiation exposures.
The following text was added in the manuscript: "The radiomic analysis utilized the venous phase from Contrast-Enhanced CT (CE-CT) scans. This phase has been noted to be more valuable for endometrial cancer staging compared to the arterial phase as it particularly highlights parenchymal characteristics, contrast dynamics, and the contrast between the tumor and myometrium (page 4 line 133-137)
References:
-Kaur, H.; Loyer, E.M.; Minami, M.; Charnsangavej, C. Patterns of Uterine Enhancement with Helical CT. Eur J Radiol 1998, 28, 250–255, doi:10.1016/S0720-048X(97)00173-3.
- Franconeri A, Fang J, Brook A, Brook OR. Asymptomatic Endometrial Thickening of 8 mm or Greater on Postcontrast Computed Tomography in Postmenopausal Women Is a Predictor of Endometrial Cancer. J Comput Assist Tomogr. 2019 Jan/Feb;43(1):136-142. doi: 10.1097/RCT.0000000000000796. PMID: 30371608.
- Lin, C.N.; Liao, Y.S.; Chen, W.C.; Wang, Y.S.; Lee, L.W. Use of Myometrium as an Internal Reference for Endometrial and Cervical Cancer on Multiphase Contrast-Enhanced MRI. PLoS One 2016, 11, doi:10.1371/journal.pone.0157820.
- Nougaret, S.; Horta, M.; Sala, E.; Lakhman, Y.; Thomassin-Naggara, I.; Kido, A.; Masselli, G.; Bharwani, N.; Sadowski, E.; Ertmer, A.; et al. Endometrial Cancer MRI Staging: Updated Guidelines of the European Society of Urogenital Radiology. Eur Radiol 2019, 29, 792–805, doi:10.1007/s00330-018-5515-y.
- Yitta, S.; Hecht, E.M.; Mausner, E. V.; Bennett, G.L. Normal or Abnormal? Demystifying Uterine and Cervical Contrast Enhancement at Multidetector CT. Radiographics 2011, 31, 647–661.
- Angelopoulos, K.; Palmér, M.; Gull, B.; Lundqvist, N.; Kopelia, L.; Hellström, M.; Leonhardt, H. Computed Tomography Contrast Enhancement Pattern of the Uterus in Premenopausal Women in Relation to Menstrual Cycle and Hormonal Contraception. Acta radiol 2021, 62, 1257–1262, doi:10.1177/0284185120958404
- Line 182. Why there were 318 EC patients for taining after data augmentation?
R: Thank you for the question and giving us the opportunity to further explain our approach. The training dataset was balanced and augmented by a factor 4. The factor “4” was chosen to reach a minimum sample size of 300 cases to improve the performance of the ML approach, according to Giesemann et al. (Reference: Julia Giesemann, Jaime Delgadillo, Brian Schwartz, Björn Bennemann & Wolfgang Lutz (2023) Predicting dropout from psychological treatment using different machine learning algorithms, resampling methods, and sample sizes, Psychotherapy Research, 33:6, 683-695, DOI: 10.1080/10503307.2022.2161432)
The following details were added in the manuscript: “The augmented and balanced dataset was created using the smoteclassif function from UBL R-package [43] which implements the SMOTE algorithms [44,45], adopting a number of nearest neighbors (i.e., k parameter) equal to 5 (corresponding to the default value), balancing the ratio of 4.4 between majority and minority classes and multiplying by an augmentation factor of 4. This value was chosen to obtain the minimum number of patients (i.e., 300) required to create the ML-based model [46].” (Page 6, lines 190-196)
Moreover, we discussed important aspects related to this approach also in the discussion section: “It is worth discussing that the SMOTE algorithm implemented for data augmentation and balancing could be affected by an over-generalization problem which could lead to the inability to capture the characteristics of a specific under-represented group of patients. This drawback can be caused by the possible oversampling of noisy or uninformative samples or the increasing overlap between different classes around the class boundaries [70]. Nevertheless, beyond the internal validation performed using a cv approach as suggested by Papanikolaou et al [71], further studies conducted on wider cohorts are needed to externally validate these results and identify radiomic markers that can be standardized and easily applied on CE-CT images collected in clinical practice in all medical centers. External validation was performed only in a limited number of studies, i.e., about an average of 6% of the cases according to a review evaluating 516 models [72] and could be further complicated by the lack of standardization among different scan machines and protocols implemented in the involved hospitals, thus affecting the radio-mic pipeline and requiring an harmonization procedure before the feature extraction in multicentric studies [73].To better generalize the proposed ML-based models, our Institute is starting a prospective multicentric study based on CT-based RF and DFS in EC patients. This paves the way for a possible future validation and adoption as a valuable ML-based tool in personalized medicine for EC patients.” (page 12 lines 419-438)
- Line 316. Table 3. What are the meaning of values? Please clarify.
R: Thank you for the question. We would like to provide here the definitions of the concepts we have used together with the adjustments of the table legend: The Feature (variable) importance indicates how much each feature contributes to the model prediction. Basically, it determines the degree of usefulness of a specific variable for a current model and prediction. The depth of a Variable (Feature) refers to how deep in a decision tree a variable is used for splits. Variables near the tree's root capture broad patterns, while those deeper capture finer details. The Occurrence of a Variable is how often a variable is chosen for splitting across all trees in the forest. High occurrence indicates strong influence on predictions, while low occurrence suggests less impact. In essence, the depth of a variable affects its contribution at different decision levels, and the occurrence of a variable signifies its importance across the entire forest. These factors collectively shape how Random Forest trees make predictions.
An explanation regarding the meaning of the variables (i.e., the radiomic features) has been added to the manuscript. Moreover, we have changed the header of Table 3 and we have added in its legend “The values represent the coefficients attributed to each feature in the ML model.” (table 3 line 343)
A: Comments on the Quality of English Language: Minor check.
R: Thank you for the observation we have further assessed the English language quality. Minor revisions have been made.
Reviewer 2 Report (Previous Reviewer 2)
The most serious problem is that this study lacks of external validation; 2. This paper lacks novelty and the analysis is not enough. 3. The sample size is too small that it cannot be published. 4. Last but not least, so many grammar errors can be found in this manuscript, please rewrite
The most serious problem is that this study lacks of external validation; 2. This paper lacks novelty and the analysis is not enough. 3. The sample size is too small that it cannot be published. 4. Last but not least, so many grammar errors can be found in this manuscript, please rewrite
Author Response
Reviewer 2
Comments and Suggestions for Authors
The most serious problem is that this study lacks of external validation; 2. This paper lacks novelty and the analysis is not enough. 3. The sample size is too small that it cannot be published. 4. Last but not least, so many grammar errors can be found in this manuscript, please rewrite
Comments on the Quality of English Language
The most serious problem is that this study lacks of external validation; 2. This paper lacks novelty and the analysis is not enough. 3. The sample size is too small that it cannot be published. 4. Last but not least, so many grammar errors can be found in this manuscript, please rewrite
R: Thank you for taking the time to provide your feedback and suggestions on our manuscript. We truly appreciate your thoughtful insights, and we have carefully considered each of your points.
External Validation: We acknowledge the importance of external validation to ensure the robustness and generalizability of our findings. While our current study is a pilot investigation focusing on the potential of radiomics in predicting disease-free survival (DFS) among endometrial cancer patients, we fully agree that future work should include a larger and diverse external validation dataset to validate our results. This will enhance the reliability and applicability of our predictive models.
Novelty and Analysis: We understand your concern regarding the novelty of our study and the depth of our analysis. We apologize if the article did not adequately convey the unique aspects of our approach. We conducted a more comprehensive literature search, but we did not find any other similar works analyzing progression-free survival linked to CT through the utilization of radiomics. Could you please kindly provide us with any references, if available?
Sample Size: We acknowledge that the sample size in our current study is limited. As you rightly pointed out, a small sample size can impact the statistical power and the generalizability of our findings. We are committed to addressing this limitation by conducting future research with a larger cohort of patients. The 'pilot study' designation was intended to indicate the preliminary nature of our investigation.
We added the following sample size estimation paragraph: We assumed a threshold probability for rejecting the null hypothesis α (two-tailed) =0.05, a power of 80%, a proportion of RF-based high-risk subjects of 0.25 and a relative hazard (high-risk Group/low-risk Group) = 0.15, thus, the total events needed are 12. Con-sidering a Baseline Event Rate (events/unit time) for low-risk Group = 0.35, a median sur-vival time in the Groups = 1.5 years, a censoring rate - censored/unit time (assumed equal for both groups) = 0 and an average length of follow-up FU = 1.5 years we calculated a sample size of 60 patients [54]. (page 6, lines 233-239)
Sample size (with continuity correction) for RF-based groups:
N Events Cumulative Event Rate
Low-risk Group 45 4 0.099
High-risk Group 15 8 0.500
Total 60 12 0.199
Reference: Schoenfeld DA. Sample-size formula for the proportional-hazards regression model. Biometrics 1983;39:499-503.
Grammar Errors: We apologize for any grammatical errors present in the manuscript. Your feedback is crucial in improving the clarity and readability of our work. We meticulously reviewed and revised the manuscript to ensure that it meets the highest standards of grammar and language.
Your valuable comments have provided us with guidance for improving our study. We are dedicated to making the necessary revisions and enhancements to address the concerns you have raised. We are grateful for your input, and we are committed to ensuring that our manuscript becomes a more comprehensive and impactful contribution to the field.
Thank you once again for your time and expertise in evaluating our work.
This manuscript is a resubmission of an earlier submission. The following is a list of the peer review reports and author responses from that submission.
Round 1
Reviewer 1 Report
The submitted work "A radiomic-based machine learning model predicts Endometrial Cancer recurrence using preoperative CT radiomic features" established a machine learning model to predict endometrial cancer recurrence using contrast-enhanced CT radiomic features. This study retrospectively enrolled 81 patients with endometrial cancer and extracted their CT radiomic features using semi-automatic segmentation approach. The patients were randomly separated into training and test sets with a ratio of 7:3, and the LASSO model was trained with 10-fold cross validation. The results showed that the predicting accuracy AUC was 0.79 (95%CI 0.66-0.92) for training set, and was 0.82 (95%CI 0.57-1) for testing set. The novelty is the first study to build machine learning prediction model using contrast-enhanced CT-based radiomic features; however, the major weaknesses are the small sample size and heterogeneous patients retrospectively enrolled for model training.
Major comments:
1) The sample size was too small for machine learning. In previous studies, the sample sizes for building a machine learning model were more than 100 cases [50-52]. An insufficient sample size and heterogeneous patient characteristics may lead to unstable prediction accuracy. Although this study showed that the AUC was as high as 0.82, the confidence interval ranged widely from 0.57 to 1.0, indicating unstable performance. So, it is needed to increase the sample size.
2) The number of patients with and without recurrence were not balanced. This imbalanced issue should be minimized by oversampling approach during the training step, so that the built model could be more stable.
3) Previous machine learning studies compared different models in search of best prediction model. It is not clear why authors only used the LASSO model to predict the recurrence. Authors are suggested to compare other models, such as random-forest, support-vector-machine, or ensemble learning.
4) This study randomly separated the datasets into training and test sets. It is correct to use random approach. However, it is not clear whether the clinical characteristics of patients were matched between the two subgroups. This should be clarified.
Minor issues:
1) Lines 20, 32, 52, 247, and 275. Some English typos should be corrected.
2) Line 148. Authors performed feature extraction and selection here, but it is not clear how feature selection was performed and how many features were selected for building a model.
3) Line 233. The uterine bodies were contoured by two observors. It is needed to understand the inter-observer reproducibility between them.
4) Line 254. After building the final model, it is needed to perform feature importance test, so as to understand what are the key radiomic features.
Some English typos should be checked and corrected.
Reviewer 2 Report
The most serious problem is that this study lacks of external validation; 2. This paper lacks novelty and the analysis is not enough. 3. The sample size is too small that it cannot be published. 4. Last but not least, so many grammar errors can be found in this manuscript, please rewrite.
many grammar errors can be found in this manuscript, please rewrite.